# Prospecting Biomarkers for Diagnostic and Therapeutic Approaches in Pythiosis

**DOI:** 10.3390/jof7060423

**Published:** 2021-05-28

**Authors:** Jéssica Luana Chechi, Tiwa Rotchanapreeda, Giselle Souza da Paz, Ana Carolina Prado, Alana Lucena Oliveira, José Cavalcante Souza Vieira, Marília Afonso Rabelo Buzalaf, Anderson Messias Rodrigues, Lucilene Delazari dos Santos, Theerapong Krajaejun, Sandra de Moraes Gimenes Bosco

**Affiliations:** 1Department of Chemical and Biological Sciences, Institute of Biosciences, São Paulo State University (UNESP), Botucatu 18618-689, Brazil; anapr1102@gmail.com (A.C.P.); cavalcante.vieira@unesp.br (J.C.S.V.); 2Department of Pathology, Faculty of Medicine Ramathibodi Hospital, Mahidol University, Bangkok 10400, Thailand; ai-kaze@hotmail.com (T.R.); mr_en@hotmail.com (T.K.); 3Faculty of Veterinary Medicine and Animal Science (FMVZ), São Paulo State University (UNESP), Botucatu 18618-681, Brazil; giselle.spaz@yahoo.com.br (G.S.d.P.); alanalucenaoliveira@gmail.com (A.L.O.); 4Department of Biological Sciences, Bauru School of Dentistry, University of São Paulo (USP), Bauru 17012-901, Brazil; mbuzalaf@fob.usp.br; 5Department of Microbiology, Immunology and Parasitology, Cell Biology Division, Federal University of São Paulo (UNIFESP), São Paulo 04023-062, Brazil; amrodrigues.amr@gmail.com; 6Center for the Study of Venoms and Venomous Animals (CEVAP), São Paulo State University (UNESP), Botucatu 18610-307, Brazil; lucilene.delazari@unesp.br; 7Graduate Program in Tropical Diseases, Botucatu Medical School (FMB), São Paulo State University (UNESP), Botucatu 18618-687, Brazil

**Keywords:** *Pythium insidiosum*, pythiosis, antigens, diagnosis, therapy, immunoproteomics

## Abstract

Pythiosis, whose etiological agent is the oomycete *Pythium insidiosum*, is a life-threatening disease that occurs mainly in tropical and subtropical countries, affecting several animal species. It is frequently found in horses in Brazil and humans in Thailand. The disease is difficult to diagnose because the pathogen’s hyphae are often misdiagnosed as mucoromycete fungi in histological sections. Additionally, there is no specific antigen to use for rapid diagnosis, the availability of which could improve the prognosis in different animal species. In this scenario, we investigated which *P. insidiosum* antigens are recognized by circulating antibodies in horses and humans with pythiosis from Brazil and Thailand, respectively, using 2D immunoblotting followed by mass spectrometry for the identification of antigens. We identified 23 protein spots, 14 recognized by pooled serum from horses and humans. Seven antigens were commonly recognized by both species, such as the heat-shock cognate 70 KDa protein, the heat-shock 70 KDa protein, glucan 1,3-beta-glucosidase, fructose-bisphosphate aldolase, serine/threonine-protein phosphatase, aconitate hydratase, and 14-3-3 protein epsilon. These results demonstrate that there are common antigens recognized by the immune responses of horses and humans, and these antigens may be studied as biomarkers for improving diagnosis and treatment.

## 1. Introduction

Pythiosis, whose etiological agent is the oomycete *Pythium insidiosum*, is an emerging and life-threatening disease that occurs most frequently in tropical and subtropical countries, affecting several animal species, especially horses, dogs, and humans [1,2,3]. This pathogen is a colonizing agent of aquatic plants in which asexual reproduction occurs via the production of sporangia. When mature, these rupture and release zoospores, the infective form of *P. insidiosum*. Zoospores move in the water until they find another plant or animal with injured tissue in which they encyst and emit a germ tube, creating a new hypha that induces the appearance of lesions [3,4].

Due to the recurrent presence in regions with water accumulation, horses are the most affected by the disease, with skin lesions being the most frequent clinical manifestation. Hyphae are observed in the lesions, covered by necrotic cells that form white-yellowish masses called “kunkers”. The intestinal form may also be present, and the animals affected have colic episodes due to obstruction of the intestinal lumen [5,6].

In humans, the disease is common in Southeast Asia, mainly in Thailand, and can also be found in the Americas and Oceania. Cutaneous and subcutaneous lesions usually take the form of granulomatous lesions or eye lesions, such as keratitis and corneal ulcers [7]. The clinical manifestation of the disease in humans can also be systemic or vascular. These manifestations are considered the most serious because they result in occlusion of the vessels [8]. In Brazil, a single report of human pythiosis was described in São Paulo, whose source of infection was due to fishing activity [9,10].

The disease is difficult to diagnose since the pathogen’s hyphae are often confused with mucoromycete fungi in histological sections, and diagnostic methods used in the identification of *P. insidiosum* are time-consuming [11,12,13,14,15]. The diagnosis of pythiosis is traditionally based on the association of clinical manifestations with culture identification [16,17,18], serodiagnostic tests [11,15,19,20,21,22,23,24,25,26], and molecular assays [13,18,27,28,29]. The treatment of pythiosis is also difficult because the pathogen does not respond satisfactorily to the available antifungals, and it is necessary to perform surgical procedures, often extensively, when possible [1,3].

Studies involving *P. insidiosum* antigens from human pythiosis have identified a 74 KDa protein, which has been considered as an immunodominant antigen [30]. This protein was identified as ß-glucanase, showing homology with a similar protein in *Phytophthora infestans*, an oomycete phylogenetically close to *P. insidiosum* [31]. In the *P. insidiosum* secretome, the elicitin antigen was analyzed in immunohistochemical tests showing 100% sensitivity in the histological sections of pythiosis tested [14,32]. The immuneproteome of *P. insidiosum* revealed antigens of different molecular weights (around 34 KDa and 50–55 KDa) that are recognized by immunoglobulins present in the serum from infected dogs, rabbits, horses, and cattle. However, serum immunoreactive proteins have not been identified [33].

The use of immunoproteomic approaches can help identify a large set of pythiosis-associated antigens that elicit immune responses during the interplay between host and pathogen. Thus, this study targeted *P. insidiosum* antigens specifically recognized by antibodies in equine and human pythiosis with the potential for use as putative biomarkers in the diagnosis and treatment of the disease. Immunoproteomic analysis was performed to select the immunoreactive molecules in the serum of these infected species.

## 2. Materials and Methods

### 2.1. Pythium Insidiosum Strain and Culture Conditions

The *Pythium insidiosum* strain used in this study, named “Eq10”, was isolated from a male horse with pythiosis from São Paulo, Brazil. This strain was identified down to the species level via DNA sequencing of ITS (Internal Transcribed Spacer) region of ribosomal DNA with the panfungal primers ITS4-ITS5 [34]. The mycelial fragments of the isolate were transferred to Erlenmeyers, containing 100 mL of Sabouraud Dextrose Broth (Difco, Detroit, MI, USA), and incubated under shaking (ES-20 Orbital Shaker-Incubator) at 120 rpm at 37 °C for five days. After that period, Sabouraud broth was removed, and the mycelial mass was washed three times with deionized sterile water.

### 2.2. Protein Sample Extraction

*Pythium insidiosum* proteins were obtained as described by Rodrigues et al., with modifications [35]. The mycelial mass was frozen in liquid nitrogen and disrupted by grinding with a pestle until a fine powder was obtained. The powder mycelium was submitted to the Precellys instrument (2 cycles of 20 s; Bertin instruments, Montigny-le-Bretonneux, France) in 1 mL of Tris–Ca^2+^ buffer (20 mM Tris–HCl pH 8.8, 2 mM CaCl_2_) containing a commercial cocktail of protease inhibitors (1:100; GE Healthcare, Chicago, IL, USA), RNase, and DNase enzymes (1:100; GE Healthcare); and glass beads (Sigma, St. Louis, MO, USA, 425–600 μm). Afterwards, the cell debris and glass beads were removed via centrifugation at 14,000× *g* for 10 min at 4 °C, and dithiothreitol (20 mM; Sigma, St. Louis, MO, USA) was added to the supernatant. The protein concentration was determined by the Bradford (Bio-Rad, Hercules, CA, USA) method [36], and the sample was kept at −80 °C.

### 2.3. Serum Samples from Infected Horses and Humans

Sera from horses with definitive diagnoses of pythiosis (via *P. insidiosum* isolation in culture) were collected during animal care at the Veterinary Hospital from the School of Veterinary Medicine and Animal Sciences, UNESP, Botucatu, São Paulo, Brazil, in 2008 and 2018 and stored at −20 °C until use. The samples (*n* = 22) were pooled to increase the recognition of seroreactive spots and avoid host-specific effects. Normal horse serum samples (*n* = 5) from animals without evidence of other mycotic diseases were used as a control.

Immunoblot was also performed with pooled serum from humans (*n* = 10) with pythiosis from Thailand. These patients had the diagnosis confirmed by culture identification, molecular analysis, and serological tests [37,38,39] at the Department of Pathology, Faculty of Medicine, Ramathibodi Hospital, Mahidol University, Bangkok, Thailand. All serum samples were stored at −20 °C until use. Serum from healthy Thai humans, who came to the Blood Bank Division, Ramathibodi Hospital, were used as control.

The use of clinical samples has been approved by the Committee for Research, Faculty of Medicine, Ramathibodi Hospital, Mahidol University, Bangkok, Thailand (assigned number: MURA2017/379), and by the Ethics Committee on the Use of Animals from the Institute of Biosciences, UNESP, Botucatu/SP, Brazil (CEUA 4345220321).

### 2.4. Two-Dimensional Gel Electrophoresis

Proteins (150 μg) were precipitated using acetone at 80% (1:4), with incubation for 90 min at 4 °C and then centrifugation at 14,000× *g* for 10 min at 4 °C. The proteins were diluted with commercial rehydration solution, DeStreak (GE Healthcare, Little Chalfont, UK), and 1% (*v*/*v*) isoelectric focusing (IEF) buffer (GE Healthcare, Chicago, IL, USA), pH 4–7, to a final volume of 125 μL. Isoelectric focusing was performed using the Ettan IPGphor III system (GE Healthcare, Chicago, IL, USA). Immobilized pH gradient (IPG) strips (pH 4–7, 7 cm) were rehydrated overnight and subsequently focused at 300 V for 1 h, 1000 V for 1 h, 5000 V for 1 h, and 5000 V for 40 min. Finally, the voltage was set to 5000 V until 7000 Vhr, all performed at 20 °C. The IPG strips were equilibrated in 19 mM dithiothreitol (DDT) in equilibration buffer (50 mM Tris-HCl pH 8.8, 6 M urea, 20% glycerol, 2% (*w*/*v*) SDS, and 0.01% (*w*/*v*) bromophenol blue) for 15 min, followed of 0.2 M iodoacetamide in equilibrium buffer for 15 min. The second dimension was performed on 12.5% (*w*/*v*) polyacrylamide gels, sealed with 0.5% agarose and proteins separated using Mini-Protean Tetra cell (Bio-Rad, Hercules, CA, USA; 150 V for 15 min and 200 V for 100 min). After separation, gels were stained with Coomassie Brilliant Blue R-250 (50% (*v*/*v*) ethanol, 5% (*v*/*v*) acetic acid) for 60 min [40], or directly transferred in the case of immunoblot analysis. The gels were scanned using the ImageScanner III (GE Healthcare Life Sciences, Uppsala, Sweden) calibrated in transmission mode.

### 2.5. 2D Immunoblotting of P. insidiosum Proteins

*Pythium insidiosum* proteins were resolved by 2D gel electrophoresis followed by immunoblotting to evaluate seroreactive spots for serum from horses and humans with pythiosis. The proteins were transferred onto a 0.2 μm nitrocellulose membrane using the Trans-Blot Turbo Transfer Pack (Bio-Rad, Hercules, CA, USA) at 25 V for 15 min using the Trans-Blot^®^ Turbo™ Blotting System (Bio-Rad, Hercules, CA, USA). Electrotransference was evaluated via Ponceau’S (0.15% Ponceau’S and 1% (*v*/*v*) acetic acid) staining.

The membranes were destained and then blocked overnight in TBST blocking buffer (20 mM Tris-HCl, pH 7.4; 137 mM NaCl; 0.1% (*v*/*v*) Tween 20, supplemented with 5% (*w*/*v*) skim milk; pH 7.4) at 4 °C under constant shaking. To perform immunoblotting from the horse serum, the membranes were probed with primary antibody diluted to 1:100 (pooled serum from horses) for 4 h at 4 °C under constant shaking. Afterwards, the membranes were washed three times with washing buffer (TBS, pH 7.5; 0.05% Tween 20) for 5 min and incubated with the secondary antibody anti-horse IgG (whole molecule)–Peroxidase, an antibody produced in rabbits (Sigma, ref. A6917, St. Louis, MO, USA), at a 1:1000 dilution for 2 h at room temperature. The membranes were washed with washing buffer, and labeling was conducted for chemiluminescent detection (Immobilon Classico Western HRP Substrate, Merck Millipore, Burlington, MA, USA) using the image analyzer Image Quant LAS 4000 (GE Healthcare Life Sciences). The immunoblotting of the Thai patient’s serum was performed with some modifications. Membranes were probed with primary antibody diluted to 1:1000 (the pooled serum of human patients) for 2 h at room temperature under constant shaking. Afterwards, the membranes were washed three times with washing buffer (TBS, pH 7.5; 0.05% Tween 20) for 5 min and incubated with secondary antibody goat anti-human immunoglobulin G conjugated with horseradish peroxidase (Bio-Rad, Hercules, CA, USA) at a 1:5000 dilution for 2 h at room temperature. Next, the membranes were washed with washing buffer, and labeling was conducted via chemiluminescence (Pierce ECL Western Blotting Substrate; Thermo Fisher, Waltham, MA USA) and image analyzer ChemiDoc^TM^ MP Imaging System (Bio-Rad, Hercules, CA, USA) [30]. The gels and membranes images were analyzed by software Image Master 2D Platinum (7.0 version, GE Healthcare, Hercules, CA, USA).

### 2.6. Trypsin Digestion and Mass Spectrometry

The 2D immunoblot seroreactive proteins were manually excised, destained with 50% (*v*/*v*) ethanol and 2.5% (*v*/*v*) acetic acid solution, dehydrated with 100% (*v*/*v*) acetonitrile (gradient grade; Merck, Darmstadt, Germany) and 50 mM ammonium bicarbonate (99.5% purity; Sigma Chemical, Barcelona, Spain), and digested with trypsin (sequencing grade; Promega, Barcelona, Spain) [41].

The analysis of the triptych peptides by mass spectrometry was performed using Ultimate 3000 LC liquid nanochromatography equipment (Dionex, Germering, Germany) coupled with Q-Exactive mass-spectrometry equipment (Thermo Fisher Scientific, Bremen, Germany). The mobile phases used were (A) 0.1% (*v*/*v*) formic acid in LCMS water and (B) 0.1% (*v*/*v*) formic acid in 80% (*v*/*v*) acetonitrile. The peptides were loaded on a C18 pre-column, 30 μm × 5 mm (Code 164649, Thermo Fisher Scientific), and desalinated in an isocratic gradient of 4% B for 3 min at a flow of 300 nL/min. Then, the peptides were fractionated by a Reprosil-Pur C18-AQ analytical column, 3 μm, 120 Å, 105 mm, Code 1PCH7515-105H354-NV (PICOCHIP) using a linear gradient of 4–55% B for 30 min, 55% at 90% B for 1 min, maintained at 90% B for 5 min and rebalanced at 4% B for 20 min at a flow of 300 nL/min. Ionization was achieved using a Nanospray ion source (PICOCHIP). The mode of operation was positive ionization using the data-dependent acquisition (DDA) method. The MS spectra were acquired from *m*/*z* 200 to *m*/*z* 2000 at a resolution of 70,000 and with a 100 ms injection time. The fragmentation chamber was conditioned with collision energy between 29% and 35% with a resolution of 17,500, 50 ms of injection time, 4.0 *m*/*z* of isolation window, and dynamic exclusion of 10 s. Spectrometry data were acquired using the Thermo Xcalibur software (version 4.0.27.19, Thermo Fisher Scientific Inc.).

### 2.7. Bioinformatic Analysis

Raw data from the mass spectrometry (RAW) were submitted to the PatternLab software (version 4.0.0.84) [42] for protein identification. The main parameters used were the SwissProt database (taxonomy *Phytophthora infestans*); trypsin enzyme; permission for 2 missing cleavages; the post-translational carbamidomethylation modification of cysteine residues; the variable post-translational modification oxidation of methionine residues; and tolerance errors of MS 40 ppm and MS/MS 0.0200 ppm. The maximum FDR (False Discovery Rate) was considered to be ≤1%.

Comparative analyses of data were performed in the Venny 2.1 program [43,44,45]. The protein–protein interaction networks were constructed using the online STRING database [46] version 11.0. The STRING network analysis was performed with a medium confidence level (0.4) [47]. The in-silico SCRATCH protein prediction suite was used to investigate antigenic properties of seroreactive proteins [48] available online [49]. The amino acid sequences of the antigens analyzed in the SCRATCH protein prediction were searched with genome information using *Pythium insidiosum* strain Pi-S (accession number GCA_001029375.1) [50]. The analyzed parameters included the determination of putative epitopic regions in antigens by evaluating continuous B-cell epitopes using COBEpro [51], the secondary structure using SSpro and SSpro8 [52], and the protein antigenicity based on multiple representations of the primary sequence using the ANTIGENpro and SVMTriP software [53].

## 3. Results

### 3.1. Pythium Insidiosum Protein Profile and Immunoreactive Proteins

The protein profile, the spots’ location, and the identification of proteins shown in our previous work [54] were consistent with the results obtained in the present study, thus demonstrating the robustness of our methodology. The protein profiles of the different *P. insidiosum* isolates studied are similar in terms of the number and molecular weight of bands and spots observed in the 1D and 2D-PAGE electrophoresis, respectively (Appendix A). Therefore, we conducted the present work using the *P. insidiosum* strain “Eq-10”.

The protein extraction protocol, Tris-Ca^2+^, was suitable for the study of *P. insidiosum* antigenic molecules, generating samples with a high quantity of proteins (2.23 μg/μL) that were not degraded (Figure 1). To evaluate the samples, 150 μg of proteins were resolved by two-dimensional gel electrophoresis. Initially, IPG strips ranging from 3 to 10 were used. The gels obtained showed the low resolution of spots in the pH range from 3 to 10 and better separation of spots by the isoelectric point using strips with a pH range of 4–7 (Appendix A). Thus, the 2D electrophoresis was conducted using an immobilized pH gradient of 4–7, obtaining better sample resolution and separation.

Two-dimensional-gel Western blotting analysis revealed a total of 23 immunoreactive spots that were immunoreactive to the pooled serum of horses and humans diagnosed with pythiosis (Figure 1). Spots with molecular weights (MWs) ranging from 28 to 88 KDa and isoelectric points (pI) between 4.58 and 6.60 were detected via the horse sera (Figure 1A,B) and human sera (Figure 1C,D). The control sera of horses and humans were not immunoreactive. 

A total of sixteen individual proteins were successfully identified by mass spectrometry from 23 immunoreactive spots (Table 1 and Appendix A). Ten spots (1–5, 9, 14, 17, 18, and 19; Figure 1) were immunoreactive for both horse and human serum. These were identified and corresponded to seven individual proteins listed in Table 2.

### 3.2. Protein Identification

The proteins resulting from the immunoblot of *P. insidiosum* with pooled horse and human serum were analyzed in the Venn diagram. Horse and human serum showed thirteen and ten proteins, respectively. Among these proteins, six (37.5%) were exclusively from horses, three (18.8%) from humans, and seven (43.8%) were shared antigens.

Network interactions among the sixteen proteins identified (Table 1) highlight the seven immunoreactive proteins matching between pooled horse and human serum (Figure 2A). Among the proteins highlighted in Figure 2A, we observed direct interactions among Hsc 70 (PITG_11913), Hsp 70 (PITG_15786), and 14-3-3 (PITG_19017). Of the sixteen proteins found in this study, eight are related to the biosynthesis of secondary metabolites and metabolic pathways, seven proteins participate in the biosynthesis of antibiotics, and six act in the pathways of amino acid biosynthesis and carbon metabolism, among other functions described in the other proteins (Figure 2B).

The predictions of the presence of B-cell epitopes showed a strong antigenicity propensity mainly in antigens Exo-1,3-ß-glucanase (antigenicity propensity: 0.935295, Figure 3A), Heat-shock cognate 70 kDa protein (antigenicity propensity: 0.899345, Figure 3B), Aconitate hydratase (antigenicity propensity: 0.895298, Figure 3C), 14-3-3 protein epsilon (antigenicity propensity: 0.892100, Figure 3D), Heat-shock 70 kDa protein (antigenicity propensity: 0.877913, Figure 3E), and Fructose-bisphosphate aldolase (antigenicity propensity: 0.738101, Figure 3F) (Figure 3, Table 2). These high antigenicity propensity scores are remarkable when compared to those for classic antigenic proteins from pathogenic fungi, such as the 43 KDa glycoprotein (AAG36672) in *Paracoccidioides brasiliensis* (antigenicity propensity: 0.604208) and the H-antigen (EGC41021) in *Histoplasma capsulatum* (antigenicity propensity: 0.777265). Serine/threonine-protein phosphatase (*Phytophthora infestans*) and anoctamin-like protein, corresponding protein in *Pythium insidiosum*, presented low antigenicity, with values of 0.345502 and 0.134275, respectively. Therefore, we did not analyze the B-cell epitopes of these proteins.

## 4. Discussion

Early diagnosis is essential for the successful treatment of pythiosis. The accurate diagnosis of pythiosis includes culture identification [16,17,18], which is time-consuming and may fail to grow the organism; molecular assays, based on PCR, require qualified personnel and specialized equipment [13,26,27,28,29]; serodiagnostic tests (e.g., immunodiffusion, enzyme-linked immunosorbent assay [ELISA], immunochromatographic tests, immunoblot, and hemagglutination) are used for the detection of circulating antibodies; however, such tests produce false-negative results [11,15,19,20,21,22,23,24,25,26]. The present study adopted an immunoproteomic approach to compare antibody responses in horse and human serum to identify new antigens in pythiosis.

The proteins extracted from *P. insidiosum* were fractionated by 2D PAGE and evaluated using Western blotting with pooled horse and human serum infected with pythiosis. Twenty-three immunoreactive spots were detected, and we identified sixteen individual proteins since the same protein was identified in two or more spots. The presence of the same protein in multiple spots can be observed due to the existence of natural isoforms, post-translational modifications, or differences in sample preparation [55].

Among the seven matching spots, the protein identification revealed heat-shock cognate 70 KDa protein (Hsc 70), heat-shock 70 KDa protein (Hsp 70), glucan 1,3-beta-glucosidase, fructose-bisphosphate aldolase, aconitate hydratase, and serine/threonine-protein phosphatase,14-3-3 protein epsilon. Heat-shock cognate 70 KDa protein was detected in three spots (1, 2, and 3), and this protein belongs to the heat-shock protein 70 family [56,57], along with heat-shock 70 KDa protein (spot 9). Hsp 70 and Hsc 70 are chaperones found in all organisms, from bacteria to humans, in the cell membrane, cytoplasm, nucleus, endoplasmic reticulum, and mitochondria, and play a critical role in various biological processes involving both systemic and cellular stress related to changes in temperature [58,59]. In *Saccharomyces cerevisiae*, a mutation in the gene Y7 caused temperature sensitivity during its growth. When analyzing the mutant yeast suppressor genes, the SSB1 gene was observed, which codes for heat-shock cognate 70 KDa protein. The introduction of this gene in mutant cells supplied the defects in temperature sensitivity and facilitated the degradation of proteins encoded by genes with the mutation [60]. Hsp 70 protein from *Paracoccidioides brasiliensis* was recognized by serum from patients with paracoccidioidomycosis [61]. Studies related to Hsp 70 and Hsc 70 proteins and their functions in oomycetes are scarce. We believe that these proteins are essential in the pathogenicity of *P. insidiosum* since the pathogen is found in the environment, and when it invades a host tissue, it needs to adapt to the higher temperature, promoting the repair and modeling of proteins fundamental to its growth.

The glucan 1,3-beta-glucosidase protein was identified in spots 2 and 3. Although this enzyme is not the most abundant protein in both spots, heat-shock cognate 70 KDa protein appears as the first identification with the highest score. However, this enzyme has been previously found in isolates of *P. insidiosum* in Brazil [54] and is also described as being immunodominant in studies conducted with serum from Thai patients with pythiosis [30,31]. Phylogenetic analysis with the *exo*1 gene, which codes for exo-1,3-ß-glucanase, grouped this gene close to other oomycetes and far from true fungi. The authors highlight this enzyme as a target for developing vaccines and drugs against the pathogen [62]. A study carried out with the pathogenic fungi *Histoplasma capsulatum* identified two proteins, endo and exo-ß-1,3-glucanase, highly active in yeasts under infection conditions, while in the mycelial form (environmental form), these proteins are minimally expressed. These proteins were related to the pathogenicity of *H. capsulatum* [63,64]. Glucanase enzyme, endo-1,3(4)-beta-glucanase, was also found as an antigen in the secretome of *Paracoccidioides* spp. (*P. americana*, *P. brasiliensis*, and *P. restrepiensis*) yeast forms [65]. The ß-glucans are the main structural components of fungal cell walls, combined with other molecules that provide strength and rigidity [66,67]. Many fungi produce exocellular enzymes that degrade ß-glucans, the ß-glucanases (ß-1,3 and ß-1,6-glucanases) [68,69]. The glucan 1,3-beta-glucosidase identified in the present study is essential in the development of *P. insidiosum* because they play a fundamental role in the morpho-physiological processes of the cell wall, thus favoring the growth of hyphae.

The mass spectrometry and bioinformatics of spots 17 and 18 resulted in the identification of the fructose-1,6-bisphosphate aldolase (Fba) protein with a molecular mass of 43 KDa and isoelectric points of 6.00 and 6.14, respectively. Fba proteins are enzymes used in glycolysis and gluconeogenesis and are classified into class I and class II [70,71,72]. Class I enzymes are present in plants, animals, algae, and prokaryotes. Class II, on the other hand, requires a metallic ion (Zn^2+^ or Fe^2+^) in the catalytic cycle. Class II is absent in animals and plants, but it is essential to protozoa, microalgae, bacteria, and fungi, favoring studies with this enzyme as targets for new drugs [73,74]. This enzyme was highlighted in a recent study as an antigen in human paracoccidioidomycosis due to *P. brasiliensis s. str.* and *P. lutzii* [75]. Moreover, proteomic data show that *P. lutzii* fructose-1,6-bisphosphate aldolase interacts with macrophage proteins. Among the proteins identified, two were identified as serine proteinase (also found in this study, spot 14, Figure 1) and Fba, both expressed on the pathogen’s surface during interaction with macrophages in the infectious process [76]. Fba was one of the *Candida albicans* proteins selected by peptide epitope search algorithms for developing a vaccine for candidiasis and showed the best results in reducing the renal fungal load [77,78]. In pathogenic fungi *Cryptococcus neoformans* [79] and *Paracoccidioides lutzii* [80], Fba binds to host molecules and performs an adhesion function, in addition to the glycolytic activity of the molecules. In this sense, we believe that the Fba protein found in our study may be involved in the interaction with host cells in the adhesion process, participating in the virulence of *P. insidiosum*.

Serine/threonine-protein phosphatase (spot 14) identified in the present study is an essential mediator of fungal proliferation and development, as well as infection-related morphogenesis and transduction. These proteins have phosphorylation functions and are essential to fundamental processes of fungi, such as cell cycle and transcription [81]. A study with *Saccharomyces cerevisiae* showed that the deletion of the SIT4 gene, which encodes the serine/threonine-protein phosphatase, increases the yeast’s sensitivity to azoles, cycloheximide, daunorubicin, rhodamine B, and rhodamine 6G, suggesting a new therapeutic approach [82]. The deletion of the gene that codes for this same protein was also analyzed in *C. albicans* [83] and *Aspergillus fumigatus* [84], resulting in more sensitivity to oxidative stress of immune cells of the host [85]. This is an important discovery since, to avoid the host’s innate immune system, pathogens need to survive oxidative conditions in neutrophils and macrophages [86].

Another detected antigen was the aconitate hydratase or aconitase (spots 4 and 5, Figure 1, Table 1). Aconitase plays a role in the Krebs cycle, catalyzing the conversion of citric acid to isocitric acid, resulting in molecules that produce energy and essential precursors for carbohydrates and amino acids [87]. In addition to its role in the Krebs cycle, this enzyme functions as an iron regulatory protein, controlling the ferritin and transferrin receptor’s expression in mammalian cells [88]. In *S. cerevisiae*, aconitase was characterized as a conjugated protein containing an iron-sulfur prosthetic group, which acts on both substrates linking to the active site or in catalysis [89]. This enzyme’s expression was evaluated in *P. brasiliensis* in the yeast and mycelial phases under cultivation in carbon sources or different levels of iron. Aconitase was more abundant in the yeast phase and when the fungus was incubated with a C_2_ carbon source or high iron concentration [87]. An increase in the expression of aconitase over a high iron concentration was also observed in *C. neoformans* [90]. The possible role of aconitase in regulating the mechanisms of iron levels is supposed [87]. Many pathogens, both prokaryotes and eukaryotes, need iron in their vital metabolic process [91,92,93]. Genetic studies with *P. insidiosum* isolates from Thailand showed the gene that encodes for the enzyme ferrochelatase [94] which is necessary for heme biosynthesis, catalyzing the introduction of iron into porphyrin [92]. Information regarding the metabolism of iron in *P. insidiosum* is scarce; however, in Thailand, an important risk factor in human pythiosis is thalassemia. Patients presenting thalassemia show an overload of circulating iron that could benefit *P. insidiosum*. Anemic rabbits experimentally infected with *P. insidiosum* showed lesions larger than those in the control group [95].

Finally, spot 19 identified the 14-3-3 protein epsilon, which is highly conserved in eukaryotes and abundant in phospho-serine/threonine binding proteins [96,97,98]. The 14-3-3 epsilon plays important roles in the cell cycle, coordinating cell progression and acting on signal transformation networks through bonds with other proteins [97,99]; these bonds affect the conformation and function of target proteins [98]. In *Paracoccidioides brasiliensis*, the 14-3-3 protein (Pb14-3-3) is a critical antigen [75], which is highly expressed in virulent isolates and works as an adhesin [100]. The downregulation of Pb14-3-3 leads to morphological modifications, such as more elongated cells, an impairment in dimorphism, decreased interaction with pneumocytes, and reduced bud number [101]. Still, in *P. brasiliensis*, Marcos et al. [102] observed that the downregulation of the 14-3-3 protein alters the pathogen’s ability to cause host cell apoptosis, which may be due to a consequence of the decreased secretion of Pb14-3-3 or the loss of putative adherence mediated by 14-3-3. In *Pythium insidiosum,* this protein may be related to the pathogen’s virulence favoring the adhesion process, as mentioned in the literature.

The proteins described here showed immunoreactivity exclusively to equine and human sera with pythiosis. There was no immunoreactivity with sera from healthy humans and equines. These antigens have already been demonstrated to be crucial in the pathogenesis and virulence of several fungi, such as *Histoplasma capsulatum* [63,64], *Paracoccidioides* spp. [61,65,75,76,80,100,102], *Aspergillus fumigatus* [84], *Saccharomyces cerevisiae* [60,82,89], *Cryptococcus neoformans* [79,90], and *Candida albicans* [83]; as well as in studies involving the development of vaccines for *Candida albicans* [77,78] and *Paracoccidioides* spp. [103]. Among these proteins, we have fructose-bisphosphate aldolase and glucan 1,3-beta-glucosidase, which are not related to homolog proteins in the human host [104].

In the present study, we have identified and characterized antigens recognized by the serum of horses and humans with pythiosis. The antigens showed here are predicted to be potent immunogenic molecules based on the presence of B-cell epitopes. These proteins can be used as new biomarkers for the diagnosis of pythiosis and treatment of this important neglected disease.

## 5. Conclusions

A total of 23 immunoreactive spots were identified for the pooled horse serum from Brazil and human serum from Thailand. The spots were identified by mass spectrometry and bioinformatic analysis, resulting in the detection of 16 unique proteins. Seven matching antigens were discussed and related to their functions in different pathogenic fungi such as glucan 1,3-ß-glucosidase, Hsc 70, Hsp 70, aconitate hydratase, serine/threonine-protein phosphatase, fructose-biphosphate aldolase, and 14-3-3 protein epsilon. The matching antigens found here may be promising for studying new methods for diagnosing and treating pythiosis.

## Figures and Tables

**Figure 1 jof-07-00423-f001:**
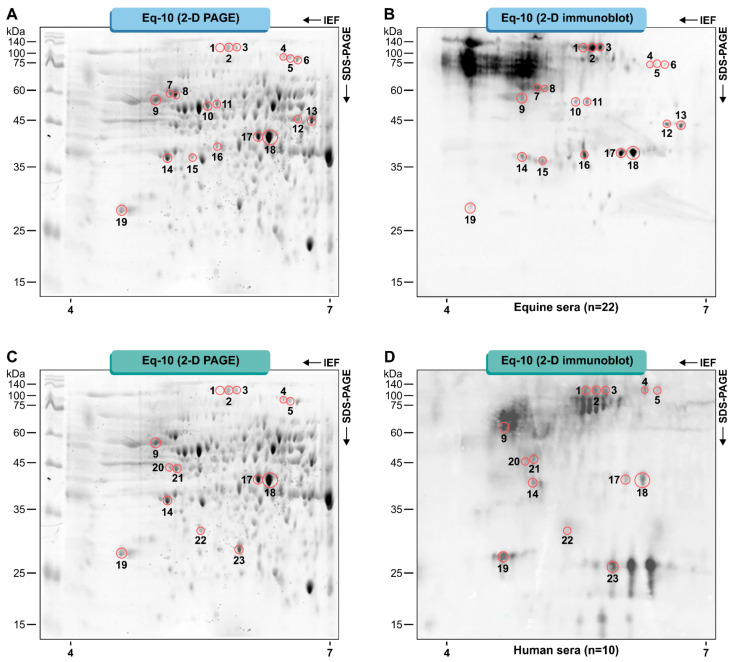
The 2D protein profile of the Eq-10 isolate of *Pythium insidiosum* (**A**,**C**). Identification of immunoreactive proteins from *P. insidiosum* by 2D Western blot analysis in pooled horse sera (**B**) and pooled human sera (**D**). The numbers of the spots refer to the identification used in Table 1 and Table 2. The experiments were run in triplicate.

**Figure 2 jof-07-00423-f002:**
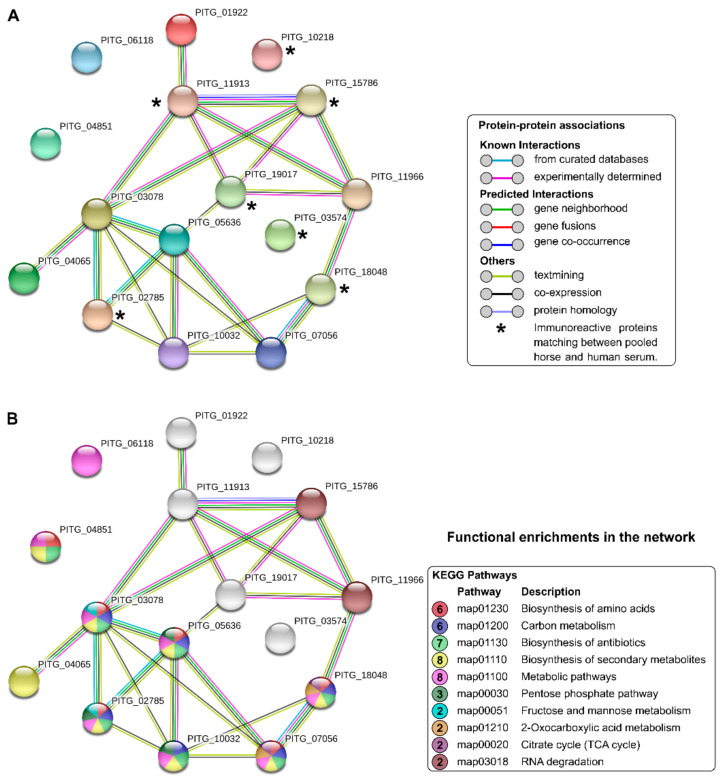
STRING analysis of the *Pythium insidiosum* immunoproteome. (**A**) The STRING protein–protein interaction network for the 16 immunogenic proteins in human and equine pythiosis. The proteins shared between the two immunoproteomes are marked with an asterisk. Colored lines between the proteins indicate the various types of interaction evidence. (**B**) Classification of proteins based on the KEGG pathways. The protein–protein interactions’ enrichment *p*-value = 3.47 × 10^−6^.

**Figure 3 jof-07-00423-f003:**
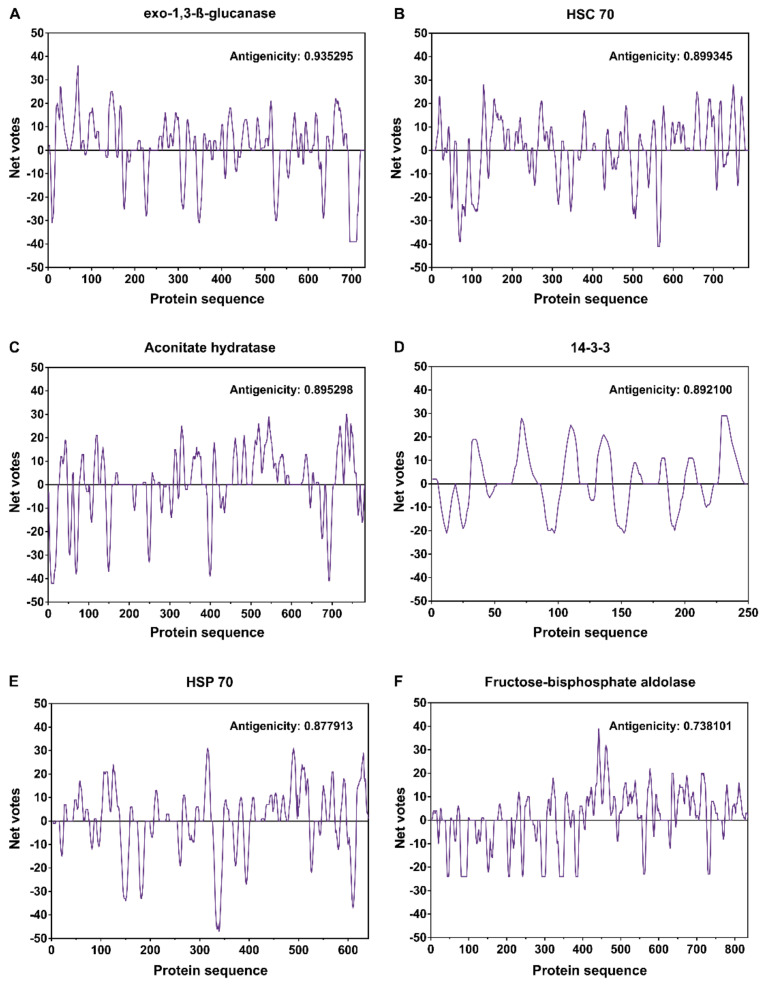
Prediction of the six antigens (**A**–**F**) of higher antigenic propensity scores. According to the algorithm COBEpro, the antigenic propensity scores are plotted against position along the amino acid sequence. The higher the antigenic propensity scores, the more likely there is to be antigenic activity for the respective region.

**Table 1 jof-07-00423-t001:** Summary of immunoreactive proteins in *Pythium insidiosum* identified by mass spectrometry.

Spot No	Protein Name	Accession No (Uniprot)	Gene	Pred MW (KDa)	pI	Protein Score	Coverage (%)
1	Heat shock cognate 70 kDa protein	D0NHI7	PITG_11913	86	5.54	66.239	16
2	Heat shock cognate 70 kDa protein	D0NHI7	PITG_11913	86	5.65	54.232	14
	Glucan 1,3-beta-glucosidase	D0NEL9	PITG_10218	69	5.65	40.664	7
3	Heat shock cognate 70 kDa protein	D0NHI7	PITG_11913	86	5.74	41.397	14
	Glucan 1,3-beta-glucosidase	D0NEL9	PITG_10218	69	5.74	17.895	5
4	Aconitate hydratase	D0NY26	PITG_18048	88	6.26	122.175	23
5	Aconitate hydratase	D0NY26	PITG_18048	88	6.35	191.649	27
6	Aconitate hydratase	D0NY26	PITG_18048	88	6.47	164.533	26
7	Chaperonin CPN60-1	D0NHM8	PITG_11966	63	5.07	293.798	34
8	Chaperonin CPN60-1	D0NHM8	PITG_11966	63	5.12	359.499	35
9	Heat shock 70 kDa protein	D0NSJ5	PITG_15786	68	4.90	12.61	8
10	6-phosphogluconate dehydrogenase	D0NE49	PITG_10032	53	5.46	82.972	28
11	Vacuolar proton pump subunit B	D0N6F5	PITG_06118	57	5.55	248.97	50
12	Isocitrate dehydrogenase [NADP]	D0N755	PITG_07056	48	6.47	188.754	43
13	Isocitrate dehydrogenase [NADP]	D0N755	PITG_07056	48	6.60	183.697	44
14	Serine/threonine-protein phosphatase	D0MXY7	PITG_03574	35	5.06	10.346	23
15	40S ribosomal protein SA	D0MUE8	PITG_01922	31	5.22	72.538	26
16	Transaldolase	D0N3B2	PITG_05636	37	5.74	24.121	16
17	Fructose-bisphosphate aldolase	D0MX78	PITG_02785	43	6.00	25.961	4
18	Fructose-bisphosphate aldolase	D0MX78	PITG_02785	43	6.14	246.075	26
19	14-3-3 protein epsilon	D0NYS2	PITG_19017	28	4.58	425.094	82
20	Arginase	D0N266	PITG_04851	38	5.00	59.674	13
21	Arginase	D0N266	PITG_04851	38	5.10	115.704	17
22	Glycerol-3-phosphate dehydrogenase [NAD(+)]	D0N0G8	PITG_04065	38	5.34	46.334	20
23	Triosephosphate isomerase	D0MZB6	PITG_03078	25	5.81	52.947	14

**Table 2 jof-07-00423-t002:** Immunoreactive protein matching between pooled horse and human serum. Predicted antigenic propensity scores based on SCRATCH and biological process from the Uniprot database are given.

Spot No	Protein	SCRATCH Score	Biological Process
2 and 3	Exo-1,3-ß-glucanase	0.935295	carbohydrate metabolic process
1, 2, and 3	HSC 70	0.899345	stress response
4 and 5	Aconitate hydratase	0.895298	tricarboxylic acid cycle
19	14-3-3	0.892100	signaling protein ligands
9	HSP 70	0.877913	protein folding and stress response
17 and 18	Fructose-bisphosphate aldolase	0.738101	glycolytic process

## Data Availability

The mass spectrometry data this manuscript have been uploaded to the MassIVE Repository from Computer Science and Engineering University of California, San Diego (ftp://massive.ucsd.edu/MSV000086565/, accessed on 1 February 2021) with the dataset identifier MSV000086565.

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
