# Peer review of "Prospecting Biomarkers for Diagnostic and Therapeutic Approaches in Pythiosis"

_jof, 2021, doi:10.3390/jof7060423_

Round 1

Reviewer 1 Report

This manuscript presents biomarkers for diagnostic and therapeutic approaches in pythiosis. While it is of interest, several flaws preclude its publication in the present form:

  1. The use of the English language leaves much to be desired. I included my remarks in the text however, they are nor exhaustive and the authors should consult someone fluent in English and/or use a grammar checking program such a Grammarly.
  2. Blood sampling for an experiment requires, at least in my country, the permit from an ethical committee. Has such a permission been obtained for sampling the horses? If so, pleas quote.
  3. Figure 1: The figure is too small and due to its limited resolution, if enlarged, becomes blurred. Please replace with a bigger, higher resolution (at least 300 dpi) version.
  4. The ratios in Figure 2 are all wrong: a circle representing 6 proteins and 3 proteins have the same size. Moreover, the area representing 7 proteins is smaller than even the 3-protein area.

Author Response

We are grateful for Reviewer 1 for the revision of our manuscript.

Considering the points raised in the review, we clarify:

1) The use of the English language leaves much to be desired. I included my remarks in the text however, they are nor exhaustive and the authors should consult someone fluent in English and/or use a grammar checking program such a Grammarly.

We submitted our manuscript for English correction, not by a native English speaker. 

2) Blood sampling for an experiment requires, at least in my country, the permit from an ethical committee. Has such a permission been obtained for sampling the horses? If so, pleas quote.

Yes, we sent our research for analysis at CEUA of our Institution. It is now mentioned in the text (see on page 3 highlighted in yellow).

3) Figure 1: The figure is too small and due to its limited resolution, if enlarged, becomes blurred. Please replace with a bigger, higher resolution (at least 300 dpi) version.

This figure was replaced in this new version.

4) The ratios in Figure 2 are all wrong: a circle representing 6 proteins and 3 proteins have the same size. Moreover, the area representing 7 proteins is smaller than even the 3-protein area.

We understand and agree with this observation, however, this figure was generated by the Venny 2.1 program (mentioned in the legend of this figure).

Reviewer 2 Report

The authors use an immunoblot/mass spectrometry approach to identify potential antigens unique to Pythium insidiosum. They were able to identify twenty-three immunoreactive spots and 16 unique proteins that may be good antigenic targets for future immunodiagnostic development studies.

Comments for the authors consideration:

1. Since most of the identified proteins are in families that are highly conserved in humans, fungi, etc (eg., heat shock proteins, B-glucans), can you provide a short discussion on how specific you anticipate these will be for P. insidiosum? Presumably they will be specific since the negative pools of human or horse serum did not display these spots but a bit more discussion of how specific you anticipate that these proteins are would be beneficial for the reader.

2. 2.2 Protein Sample Extraction, first sentence: Pythium insidiosum proteins were …. (were replacing was);

3. 2.4 Two-dimensional gel electrophoresis, fifth sentence: dithiothreitol should not be capitalized;

4. 2.6 Trypsin digestion, paragraph 2: please define "DDA" in DDA method or provide a reference;

5. 2.7 Bioinformatics, paragraph 2 - Pythium insidiosum should be italicized;

Author Response

We are grateful for Reviewer 2 for the revision of our manuscript.

Considering the points raised in the review, we clarify:

1. Since most of the identified proteins are in families that are highly conserved in humans, fungi, etc (eg., heat shock proteins, B-glucans), can you provide a short discussion on how specific you anticipate these will be for P. insidiosum? Presumably they will be specific since the negative pools of human or horse serum did not display these spots but a bit more discussion of how specific you anticipate that these proteins are would be beneficial for the reader.

A new paragraph was introduced (see on page 13, highlighted in yellow).

2. 2.2 Protein Sample Extraction, first sentence: Pythium insidiosum proteins were …. (were replacing was);

It was corrected.

3. 2.4 Two-dimensional gel electrophoresis, fifth sentence: dithiothreitol should not be capitalized;

It was corrected.

4. 2.6 Trypsin digestion, paragraph 2: please define "DDA" in DDA method or provide a reference;

It was corrected.

5. 2.7 Bioinformatics, paragraph 2 - Pythium insidiosum should be italicized;

It was corrected.

Round 2

Reviewer 1 Report

The role of graphs, in my oppinion, is to add a visual aspect to data presented in text form, especially if the latter is complex. In this case, figure 2 does not add significantly to the text and, as I mentioned in my previous review, does not represent it visually. Thus I suggest removing it alltogether.

Author Response

Dear reviewer,

We thank for your suggestion and we removed this figure in the present version of the manuscript follow attached.
